# Using Palaeomagnetic Techniques to Date Indigenous Archaeological Sites in New Zealand

Shefali Poojary, Fergus Robinson  and Gillian Turner *

School of Chemical and Physical Sciences, Te Herenga Waka–Victoria University of Wellington, P.O. Box 600, Wellington 6012, New Zealand; shefalipoojary3012@gmail.com (S.P.); fergus123robinson@gmail.com (F.R.)
* Correspondence: gillian.turner@vuw.ac.nz

**Abstract:** Aotearoa/New Zealand was first settled by the Māori people some 800–1000 years ago. Archaeomagnetism provides one of the few means of dating early sites of Māori occupation, particularly when radiocarbon dating is not feasible. This involves dating the thermoremanent magnetization imparted to the heat-retaining stones used in traditional Māori earth ovens, hāngī or umu, at the time of their last cooling. The direction of this magnetization is correlated with the reference curve of the changes in the geomagnetic field direction in New Zealand over the past 1000 years, NZPSV1k.2023. Here, we describe the application of archaeomagnetic dating to indigenous hāngī sampled at two sites in the North Island of New Zealand. The first, in the present-day city of Napier on the east coast, has been studied in detail and is shown to have been occupied, possibly intermittently, over 400–600 years, while the second, in present-day Waikanae on the west coast, is tentatively dated to ca. 1760 AD, just decades before the first European arrival in New Zealand.

**Keywords:** Palaeomagnetism; archaeomagnetic dating; Aotearoa/New Zealand; Māori; earth ovens (hāngī); hāngī stones

## 1. Introduction

New Zealand/Aotearoa is the most recently settled land mass on the globe—the great waka (two-hulled, ocean-going canoes) having arrived from east Polynesia probably only 900–800 years ago [1]. Early Māori settlements spread quickly around the coasts of both North and South Islands, where bush was cleared from modest areas of land to make way for villages and gardens, and where kai moana (seafood) was plentiful. Indigenous technology was very simple, with timber being used for building and cooking carried out in umu or hāngī (earth ovens), with the help of heat retaining stones (Figure 1). History was passed down orally from generation to generation, with very little being recorded prior to the arrival of Europeans in 1770 AD. Subsequently, unrest between Māori tribes (iwi) became more commonplace, as well as disputes and battles between Māori and the European colonists [1].

In many parts of the world the remanent magnetization of fired archaeological artefacts or materials has been used as a means of age estimation [2]. Most natural rocks, sediments and the clays used in pottery and brick-making contain small percentages of ferro or ferri-magnetic minerals, such as magnetite or haematite [3]. In addition, the firing or heating process may cause thermochemical alteration of clay minerals resulting in enhanced levels of magnetite [4]. Such minerals acquire a thermoremanent magnetization (TRM) on cooling through their characteristic Curie temperatures (585 °C for magnetite and 680 °C for haematite; lower for titanomagnetites or titanohaematites depending on titanium content [3]). This TRM is generally parallel to the ambient magnetic field in which the last cooling occurs, and its strength is proportional to the strength or (palaeo) intensity of the field [3].

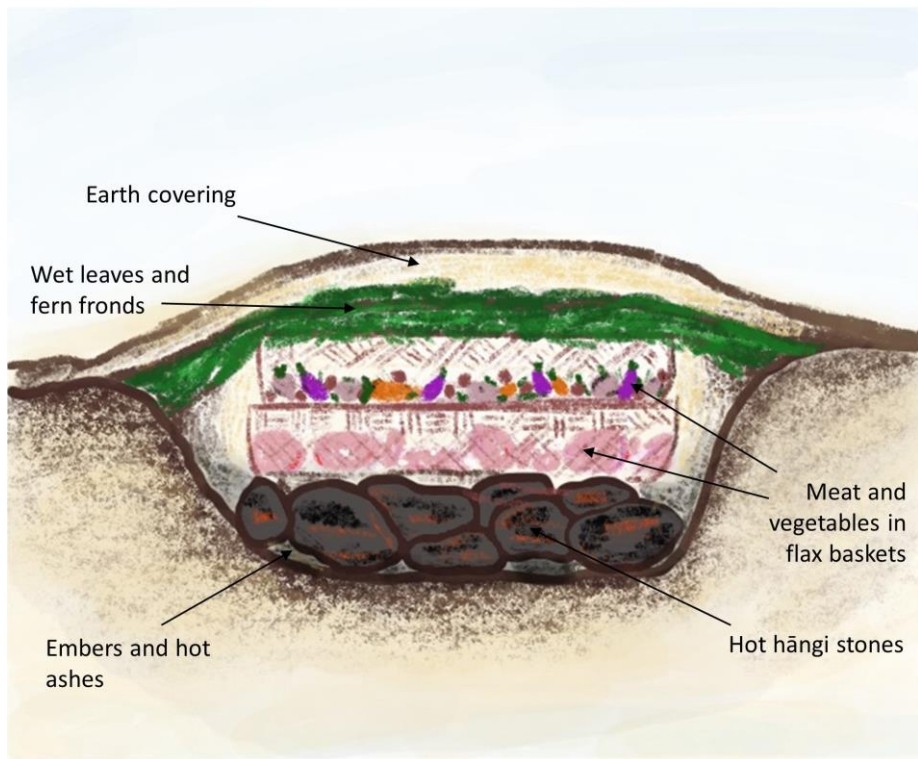

**Figure 1.** A traditional Māori hāngī (earth oven). A hāngī pit may be 0.5–1 m in depth, and 0.5–2 m in diameter, depending on the amount of food to be cooked.

Although roughly resembling the field of a geocentric axial dipole (GAD), in detail, the geomagnetic field is spatially complex and varies with time. At any location and point in time, its direction can differ from the GAD direction by up to a few tens of degrees. The temporal variation of the geomagnetic field is known as (palaeo) secular variation (PSV). PSV is typically coherent over regions approximately 1000–2000 km in extent, for which it is possible to develop reference curves and/or regional models.

By correlating the direction and/or palaeointensity extracted from an archaeological artefact with a regional PSV reference curve, it is possible to estimate the age of the TRM, and so date the last cooling of the artefact. Archaeomagnetic studies are well-developed in parts of the world which have rich cultural histories and plentiful archaeological materials: for instance, the UK and Europe, e.g., [5–7], the Mediterranean, e.g., [8], China, e.g., [9] and Africa, e.g., [10].

In this study, we focus on hāngī stones from New Zealand, using archaeomagnetism to date some of the early sites of occupation around the country [11–13]. To retain a useful record of the geomagnetic field direction, hāngī stones must have remained undisturbed in the positions and orientations in which they last cooled. An additional challenge of working with such discrete samples is therefore that of identifying the records of stones that have remained in situ and discarding those that have been disturbed [14]. We present two case studies, one from Napier/Ahuriri on the east coast and one, Ngārara, on the west coast of the North Island (Figure 2).

### 1.1. Ahuriri/Napier

Napier is a port city in the Hawkes Bay region of the east coast of the North Island (Figure 2). In 1931, the area suffered a magnitude 7.8 earthquake, which destroyed much of the growing city, and caused significant uplift of the nearby land. Prior to the earthquake, Napier was centred on a rocky hill (Scinde Island, later renamed Bluff Hill), with much of its surroundings being lagoons, tidal swamps or salt marshes (Figure 3a). The earthquake lifted and drained Ahuriri Lagoon to the northwest of the city, as well as much of the foreshore

area to the south of Bluff Hill, creating some 2000 hectares of new land [15,16]. Figure 3b is a contemporary map showing today's airport northwest of the city and the spreading urban development to the south and southwest. The site described here, at 39.49° S, 176.91° (marked with a star), was exposed during the demolition of post-earthquake houses to make way for a modern office building. It lies almost immediately south of Bluff Hill, on what would, prior to the earthquake, have been the edge of a swamp.

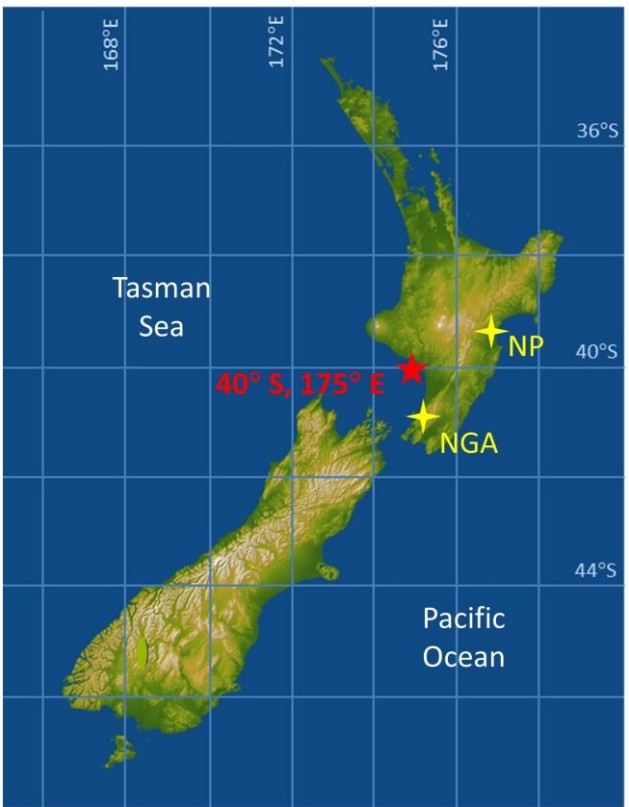

**Figure 2.** Map of New Zealand showing the locations of archaeological sites, NP (39.49° S, 176.91° E) and NGA (40.87° S, 175.04° E), and the palaeomagnetic reference location, 40° S, 175° E.

Covering an area of about 2000 m², the site exposed a natural gravel/sand/shell bar some 50–75 cm below current ground level, which was covered in shells and small fragments of pumice (Figures 3 and 4a). Excavation uncovered two well-preserved hāngī, along with several smaller cultural features, including stone-lined hearths and miscellaneous artefacts indicating pre-European indigenous occupation. The site is catalogued as V21/504 in ArchSite, the database of the New Zealand Archaeological Association (NZAA), last accessed 17 September 2023. Stones were oriented and collected from the two major hāngī, NP1 (Figure 5a) and NP2, and the baked floor of a fireplace (NP5, Figure 5c) was sampled in 24 × 24 × 20 mm clear plastic boxes.

*1.2. Ngārara, Kapiti Coast*

The second site, Ngārara (40.87° S, 175.04° E), is at Waikanae on the Kapiti Coast some 55 km north of the capital, Wellington (Figure 2). It lies in an extensive belt of dune sands that date back several thousand years and continue to accumulate at the coast today [16]. Archaeological work in the area documents early temporary settlements around stream and river mouths and more permanent villages further inland [17]. At the beginning of the nineteenth century, the area was home to the Muaūpoko and Ngāti Apa people. However, soon after this groups moved south from Taranaki and Waikato, leading to unrest and the so-called "musket wars", an influx of European settlers and a decline in the local Māori population. The site described here was uncovered during earthworks for a new retirement

village, during which numerous features were discovered [18,19]. Many other, possibly related features were discovered and described during construction of the nearby Kapiti Expressway [19].

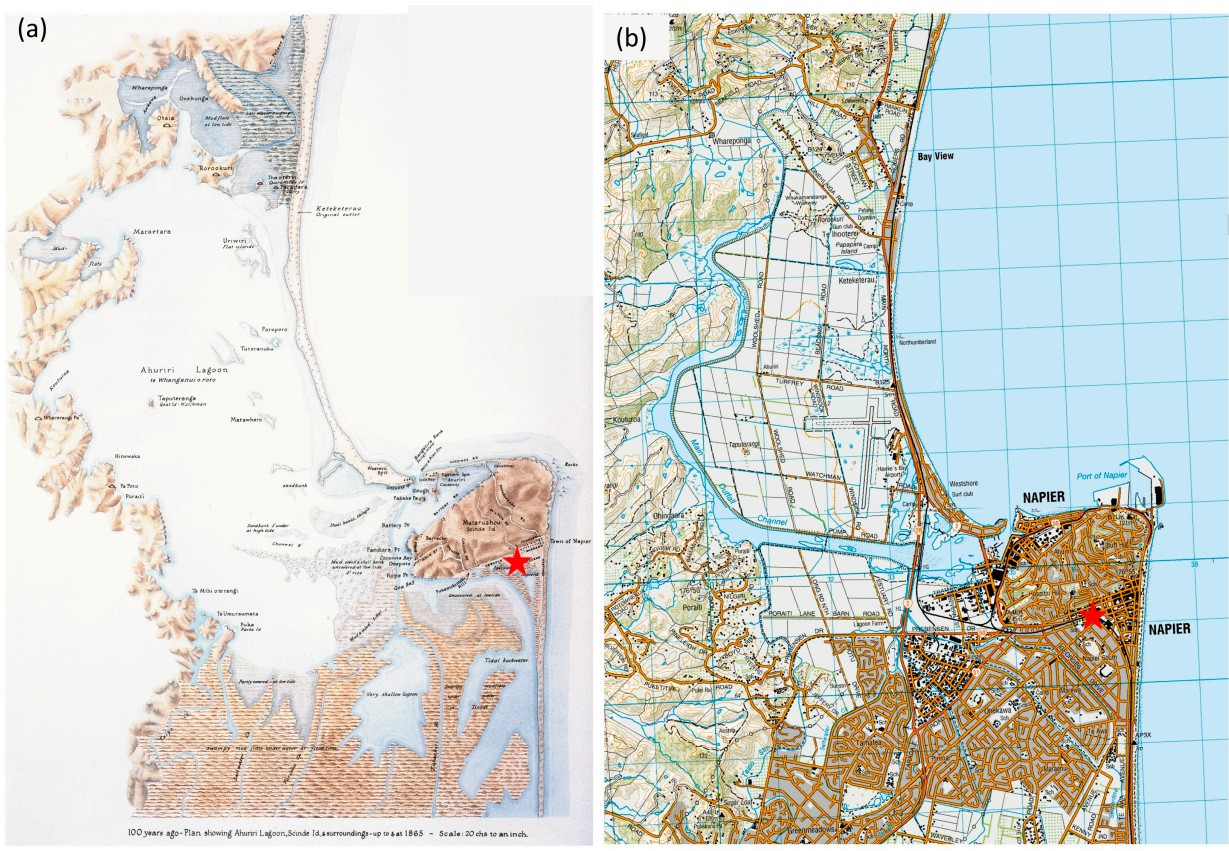

**Figure 3.** (**a**) Historical map of the Ahuriri/Napier area made ca. 1865 AD. Courtesy of MTG Hawke's Bay (Napier museum). (**b**) Contemporary topographic map of the same area (www.topomap.co.nz, accessed on 6 August 2023). Grid squares are 1 km × 1 km. The red stars mark the archaeological study site. Napier city centres on Bluff Hill which, in 1865, was known as Scinde Island or Mataruahou. In 1865, the island was surrounded by lagoons, tidal backwaters and swamps. The 1931, Napier earthquake raised these areas above sea-level, enabling urban development to the south, and the airport and agricultural development to the north.

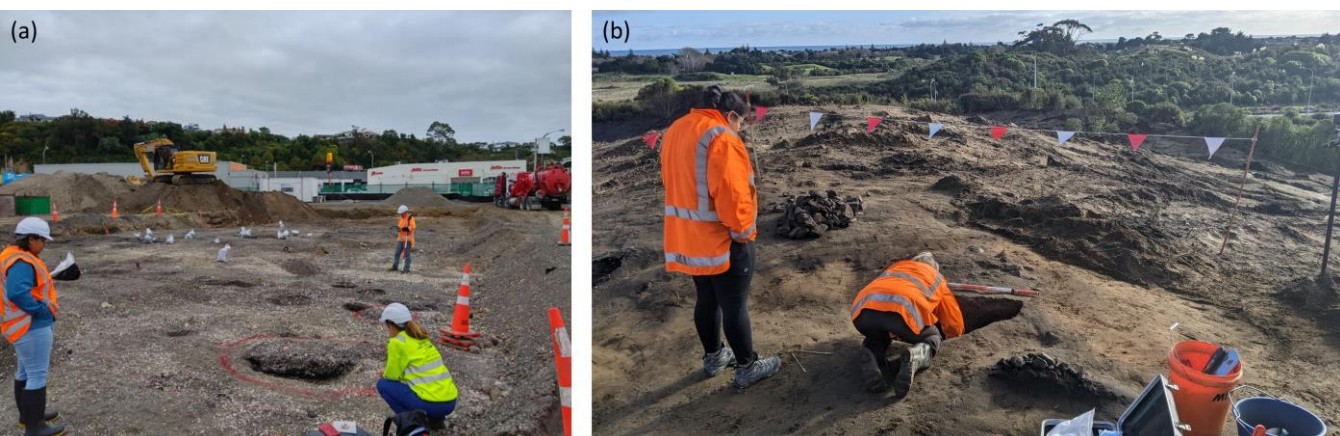

**Figure 4.** (**a**) Overview of the Napier archaeological site, hāngī NP1 and NP2 are in the foreground; (**b**) the Ngārara (Waikanae) archaeological site with hāngī NGA1 under excavation, looking west across the Kapiti expressway and sand dunes towards the coast.

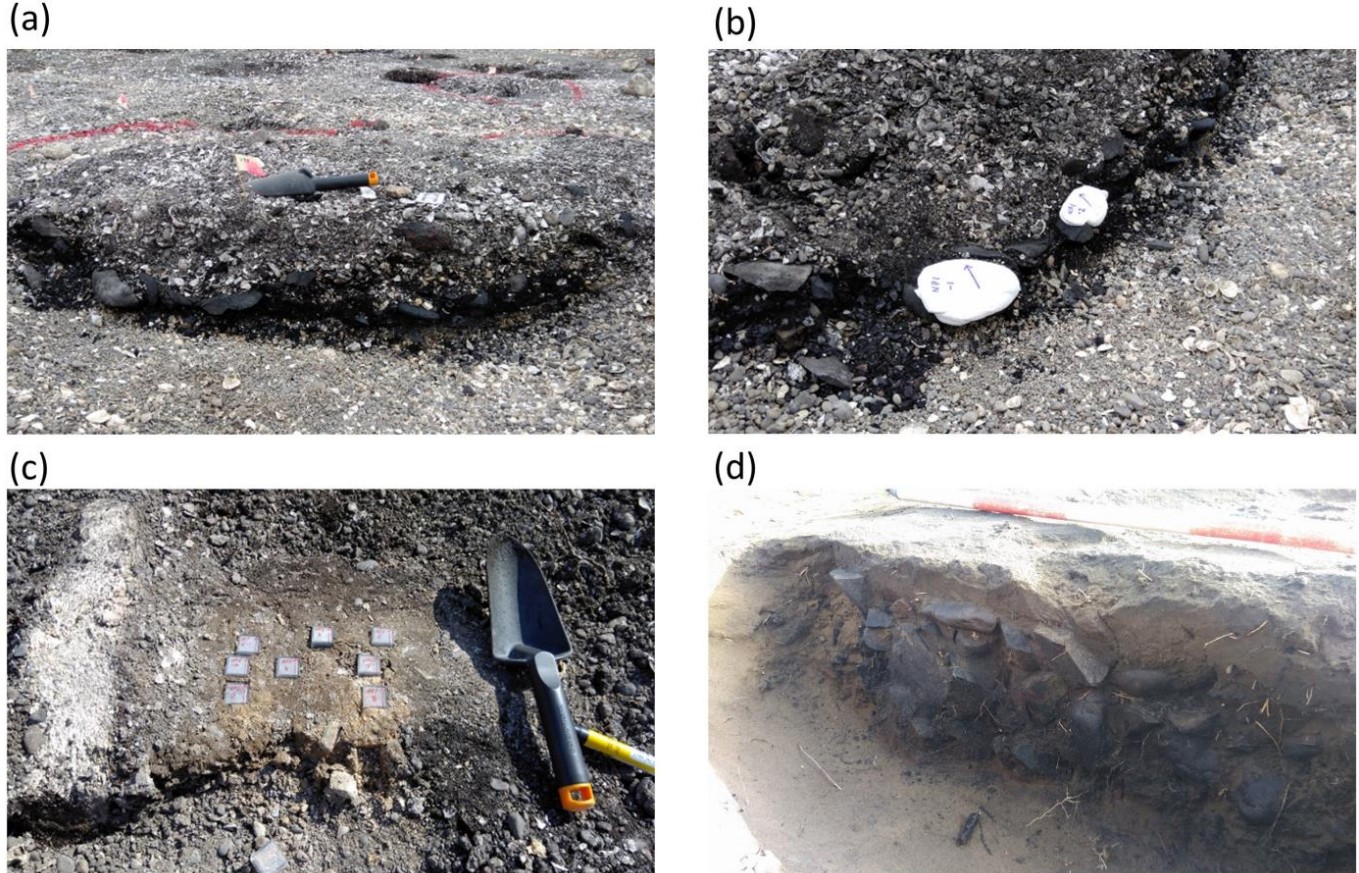

**Figure 5.** (**a**) Hāngī NP1; (**b**) sampling method stones NP1-1 and NP1-2; (**c**) baked hearth-floor, NP5; (**d**) Hāngī NGA1, half sectioned, before sampling. The plastic trowel in (**a**,**c**) is 30 cm long.

Here, we describe preliminary results obtained from stones sampled from one large hāngī, NGA1, excavated at the Ngārara site (Figures 4b and 5d). This hāngī is the biggest and best-preserved of a cluster of four ovens and other features recorded in the NZAA ArchSite database as R26/849. It is given the label NGA088 in archaeological site records. Although no material suitable for radiocarbon dating was recovered from NGA1/NGA088 itself, a radiocarbon age of 319 ± 17 C-14 yr BP (Wk-56748) has been obtained from a nearby midden (NGA089), which is also part of the cluster of features listed as R26/849, and might be contemporaneous with NGA1/NGA088. This age has been calibrated to a range in calendar years between 1500 and 1660 AD, using the SHCal20 calibration curve [20].

### 1.3. Māori Hāngī Practice

A hāngī is "laid down" by first building a fire within or above a 0.5 to 1 metre deep pit, and interspersing stones with the firewood before igniting. Māori have always preferred volcanic rocks, for their resistance to cracking under intense heat; however, in coastal areas far from the central North Island volcanoes, the local greywacke, washed out of the mountain ranges and found in riverbeds, was often the only option. Once the fire is lit, the stones begin to absorb heat—they can easily reach temperatures of 700–800 °C or higher—and as the wood burns, the stones fall into the bottom of the pit. With hot stones in the pit and the fire essentially burned out, baskets of food and damp vegetation are layered on top. The whole is finally topped with an insulating layer of vegetation and soil and left for several hours for the food to steam-cook (Figure 1). When the hāngī food is eventually lifted, the stones are left in place to cool. Sometimes the site is covered and abandoned, sometimes the stones are cached for reuse. It is the abandoned sites that are of value for archaeomagnetic dating, if it can be ascertained that (some of) the stones have remained in

situ since their last cooling, and so retain a TRM record of the geomagnetic field direction (and/or intensity) at the time.

In an experimental hāngī, laid down at Waiwhetū Marae, just north of Wellington, during the Māori midwinter festival of Matariki in 2012, it was shown that hāngī stones can reach up to 1000 °C, and cool only very slowly once the hāngī food has been laid on them and covered to cook. A variety of stones from around New Zealand was included in the hāngī. After orienting and retrieving the stones, they were sampled, and their remanent magnetization studied. It was found that they had acquired strong TRMs, which grouped tightly around the local magnetic field direction [21].

## 2. Methods

### 2.1. Sampling

To sample a stone, first its upper surface must be carefully exposed and cleaned as well as possible without disturbing it. Then, a Plaster of Paris cap is applied and is levelled horizontally using two perpendicular spirit levels. Once the plaster is set, magnetic north is marked on it, as well as the bearing to the Sun (if possible), and the precise time is noted (Figure 5b). This gives two independent means of azimuthal orientation, which can later be checked against each other. The stone is then carefully removed, given a unique label, and is wrapped for transport to the laboratory. Standard practice is to collect between ten and twelve stones from different levels in a hāngī, keeping note of their location in the feature.

Once in the laboratory, the stones are set upside down in wooden boxes using expanding foam, so that standard palaeomagnetic core samples, one inch (25 mm) or half an inch (12.5 mm) in diameter, can be drilled from them, while accurately maintaining their orientation. The core samples are subsequently cut into 22 mm or 11 mm long specimens.

NP5 was the baked base of a fireplace. Although the material was somewhat sandy and poorly consolidated, it was sampled in order to trial a method of direct sampling of undisturbed baked material. Clear plastic boxes with dimensions of approximately $24 \times 24 \times 20$ mm were pushed vertically into the baked surface, which had been levelled to horizontal (Figure 5c). The samples were oriented with a magnetic compass, labelled, carefully removed and sealed with a lid, before packing for transport to the laboratory.

Table 1 gives details of the stones and samples obtained from each site and feature.

**Table 1.** Archaeological sites and features studied and details of samples collected.

| Site/Feature | Lat (°S), Long (°E) | Grid Ref (NZTM2000) | Arch-Site No. | Feature | Date Sampled | Samples Collected | Specimens Analysed |
|---|---|---|---|---|---|---|---|
| **Napier** | | | | | | | |
| **NP1** | 39.49, 176.91 | E1936301 N5621098 | V21/504 | hāngī/earth oven | 16/01/2020 | 11 oriented stones, NP1-1 to NP1-11 | 49 (most THD) |
| **NP2** | 39.49, 176.91 | E1936301 N5621098 | V21/504 | hāngī/earth oven | 16/01/2020 | 11 oriented stones, NP2-1 to NP2-11 | 33 (most THD) |
| **NP5** | 39.49, 176.91 | E1936301 N5621098 | V21/504 | baked floor of oven/fireplace | 16/01/2020 | 8 oriented samples in plastic boxes, NP5-1 to NP5-8 | 6 (AFD) |
| **Ngarara** | | | | | | | |
| **NGA1** | 40.87, 175.04 | E1772070 N5473987 | R26/849 | hāngī/earth oven | 30/06/2022 | 11 oriented stones, NGA1-1 to NGA1-11 | 17 (AFD) |

### 2.2. Magnetic Measurements

Magnetic remanence measurements were made using an Agico JR-6 spinner magnetometer, operated through Rema6W software, which computes and saves the declination, inclination and intensity of the magnetic remanence in both the "specimen" and geographic reference coordinates. Progressive thermal demagnetization (THD [3]) was carried out on

most NP1 and NP2 specimens in a Magnetic Measurements shielded oven, while alternating field demagnetization (AFD [3]) was carried out on others, including all specimens from NP5 and NGA using a Molspin demagnetizer. Specimens are placed in the demagnetizers in opposite orientations between successive demagnetization steps to compensate for any undetected residual magnetic fields in the equipment. Principal Component Analysis (PCA [22]) was conducted on the demagnetization data in Agico's Remasoft application [23] to obtain the direction of the required "characteristic" component of the TRM (the ChRM). Remasoft was also used to plot data and for calculation of stone and site mean ChRM directions and statistical parameters.

### 2.3. Archaeomagnetic Dating

Once a mean palaeomagnetic ChRM direction has been obtained for an archaeological feature, together with a measure of confidence, usually $\alpha_{95}$, the semi-angle of the cone of 95% confidence in the mean [24], this can be compared with a reference palaeosecular variation record for the region. The best match can be used to provide an estimate of the date of the feature. In practice, the MATLAB "archaeomagnetic dating tool" of Pavón-Carrasco et al. [25] was used, together with NZPSV1k.2023, the high-resolution reference curves for New Zealand covering the past millennium [26,27]. The dating tool returns probability density functions for each of declination, inclination and intensity (if available), and an overall probability density function for the match. The result may be calculated at 65% or 95% confidence levels. NZPSV1k.2023 is compiled from geomagnetic observatory records covering the past century (Intermagnet.org: international real-time geomagnetic observatory network), from the global geomagnetic model, gufm1 [28], which reaches back to 1600 AD, and from a detailed compilation of dated lake sediment records for the preceding 600 years. NZPSV1k.2023 is calculated at 40° S, 175° E and all input data are relocated to this reference site. Dates are given in calendar years AD (CE).

## 3. Results

### 3.1. Napier

Sampling details, methods and results from the Napier archaeological site are discussed in detail in the MSc thesis of Poojary (2023 [12]). Poojary [12] also carried out detailed rock magnetic studies showing that the TRM was carried by fine-grained (single domain) iron-rich titanomagnetite, probably thermo-chemically enriched in concentration during the hāngī heating process.

Eleven stones, most of which were 10–20 cm along their longest dimension, were sampled from the biggest hāngī, NP1, and between two and ten specimens were analysed from each stone. NRM intensities ranged between 0.1 and 6.1 A/m. Figure 6a,b show progressive demagnetization data for two representative specimens. NP1-10-E1 is an example of a stone that has remained in situ throughout and since cooling. It carries a single component of thermoremanent magnetization, which on PCA, gives a best ChRM direction with declination, Dec = 334.8°, inclination, Inc = −50.9° and Maximum Angular Deviation, MAD, (a measure of goodness of fit, smaller being better) of 0.7°. Specimen NP1-5-B1, on the other hand, shows two components of TRM. The component removed at heating steps up to 500 °C is in a similar direction to the TRM of NP1-10-E1, and is interpreted to reflect the local magnetic field at the time of cooling. The component removed at higher temperatures, however, has an opposite, downward vertical component. This suggests that, after cooling to about 500 °C, the stone was disturbed and rotated into its final orientation, in which it subsequently cooled to ambient temperature and remained until sampled. The low blocking temperature component, acquired between 500 °C and ambient temperature, is therefore the ChRM of interest in this study. In Figure 7a, the ChRM directions of 27 specimens from seven NP1 stones are plotted, together with their mean direction and 95% confidence limit ($\alpha_{95}$). Data from the remaining four stones were excluded, since their mean directions fell beyond two circular standard deviations (28.8°) of the overall mean, indicating that the stones had been disturbed after cooling and so were

no longer in situ. Data from a further two specimens were excluded, as they fell outside two circular standard deviations of their stone mean directions. The most common reasons for this include the ChRM residing in a very small portion of the blocking temperature spectrum, and so being poorly resolved, or orientation errors during sample preparation. The complete dataset is given in the appendix. The overall mean ChRM direction for hāngī NP1 is Dec = 353.0°, Inc = −63.1°, $\alpha_{95}$ = 4.8°, N = 27 specimens (Table 2).

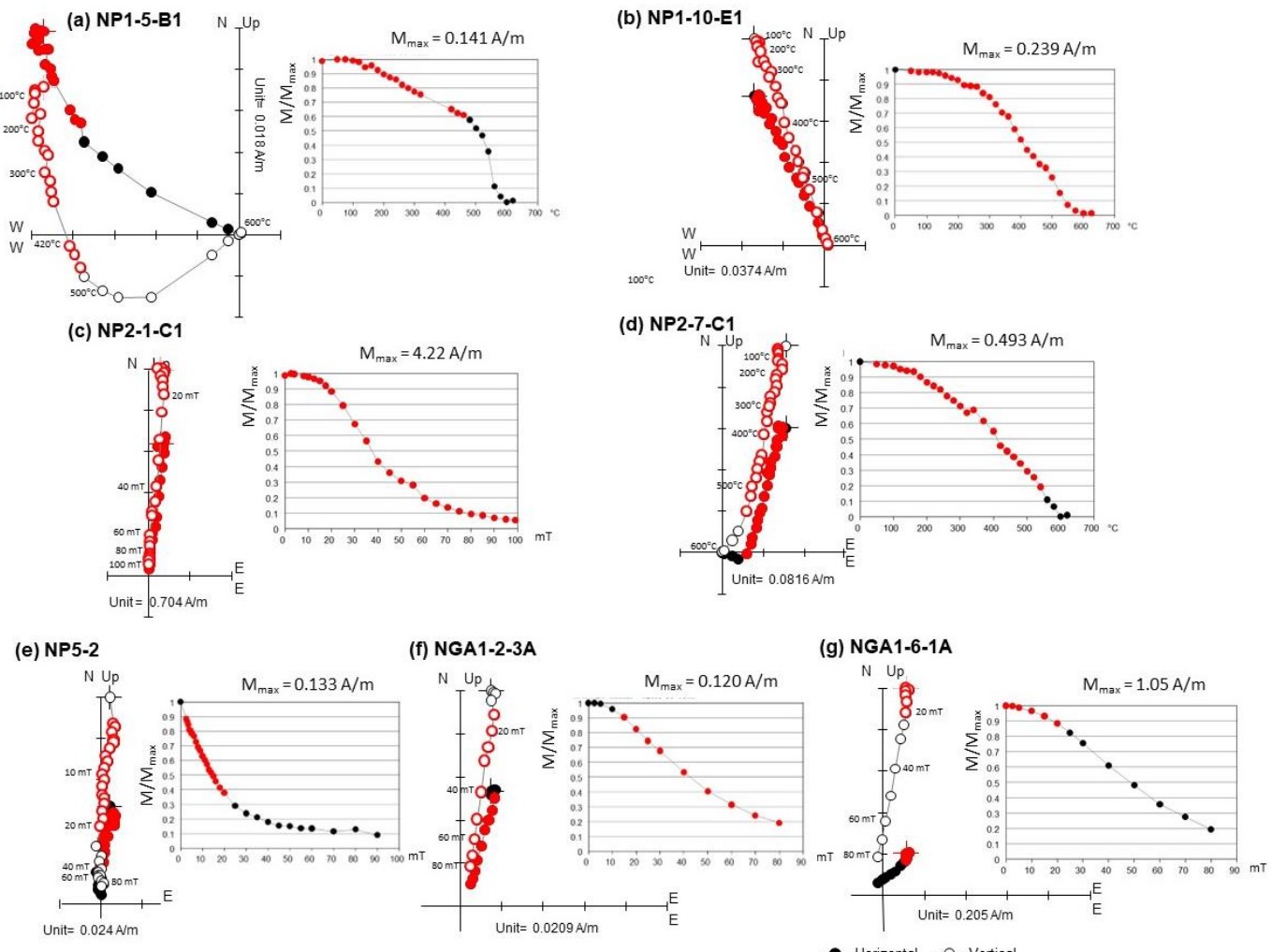

**Figure 6.** Vector component plots and plots of normalised intensity reduction during progressive demagnetization for selected specimens. (**a**) NGA1-2-3A; (**b**) NGA1-6-1A; (**c**) NP1-10-E1; (**d**) NP1-5-B1; (**e**) NP2-1-C1; (**f**) NP2-7-C1; (**g**) NP5-2. The red symbols denote the datapoints that define the ChRM, which were selected for principal component analysis. On vector component plots the solid symbols denote the horizontal (N vs. E) component and open symbols denote the vertical (upward vs. E) component. Demagnetization steps (temperature or peak alternating field) are shown on the vertical component data.

Eleven stones were sampled from hāngī NP2, with a total of 33 specimens analysed and having NRM intensities between 0.5 and 11 A/m. Specimen NP2-1-C1 (Figure 6c) displays typical single component TRM demagnetization data from this hāngī. In contrast to data from NP1, the horizontal component is noticeably to the east of north, with the best PCA direction for this specimen having Dec = 6.2°, Inc = −56.3°, MAD = 1.0°. Specimen NP2-7-C1 returns a similar direction, but has a small divergent high blocking temperature component, indicating disturbance of stone NP2-7 at about 550 °C, before settling into its final orientation. Data from 18 specimens from seven stones were included in calculation

of the mean ChRM for hāngī NP2: Dec = 7.6°, Inc = −56.9°, $\alpha_{95}$ = 4.1° (Figure 7b, Table 2 and Appendix A).

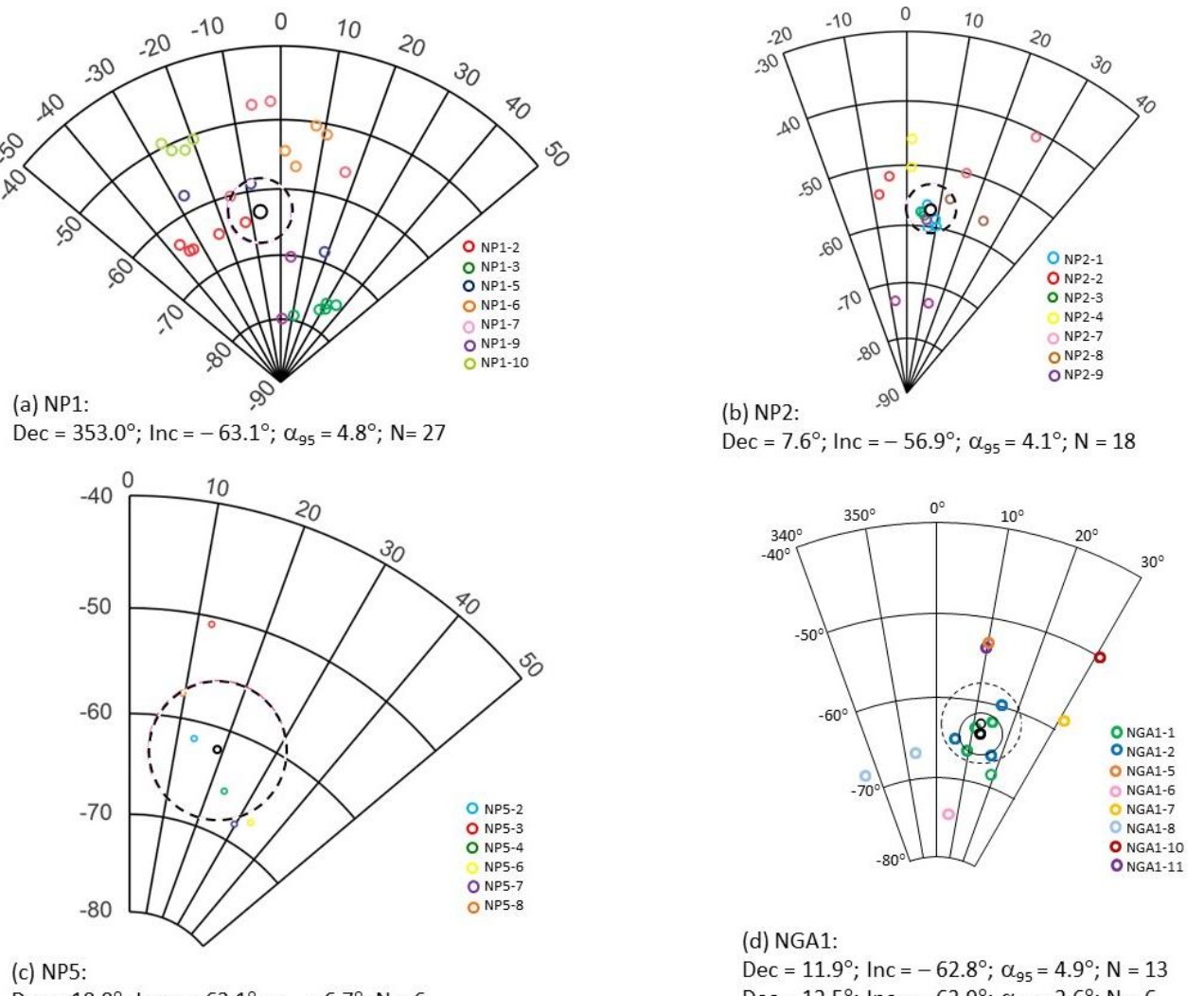

**Figure 7.** Partial stereographic (Wulff) projections showing specimen ChRM directions used to calculate mean directions for each feature; mean ChRM directions and 95% confidence limits ($\alpha_{95}$) in black (**a**) NP1. Data from stones NP1-1, NP1-4, NP1-8 and NP1-11 were disregarded since outside $2\theta_{63}$; (**b**) NP2. Data from stones NP2-5, NP2-6, NP2-10 and NP2-11 were disregarded since outside $2\theta_{63}$; (**c**) NP5. Data from specimens NP5-1 and NP5-5 were disregarded as magnetization was very unstable; (**d**) NGA1. Stone NGA1-3 was not sub-sampled. Data from stones NGA1-4 and NGA1-9 were disregarded since outside $2\theta_{63}$. Two estimates of the NGA1 mean ChRM direction are shown. See text and Appendix A for full data and details.

Six specimens from NP5, the baked fireplace floor, were analysed. The mean NRM intensity, approximately 0.1 A/m decreased rapidly with AFD, with median destructive fields between 10 and 15 mT. Demagnetization data from a typical specimen is shown in Figure 6e, returning a ChRM direction, Dec = 13.5°, Inc = −61.7°, MAD = 2.2°. In general, the MADs for NP5 specimens were higher than for specimens from hāngī stones, probably due to grain movement within the largely unconsolidated sandy material. The mean ChRM direction from the six specimens has Dec = 18.8°, Inc = −62.1°, $\alpha_{95}$ = 6.7°. (Figure 7c, Table 2 and Appendix A).

**Table 2.** Feature-mean ChRM directions, and best estimates of archaeomagnetic dates of each feature. N = number of specimen data included in ChRM of $N_0$ specimens analysed. n = number of stones from which successful data were obtained of $n_0$ stones studied.

| Site/Feature | N/$N_0$ (Specimens) | n/$n_0$ (Stones) | Dec (°) | Inc (°) | $\alpha_{95}$ (°) | Archaeomagnetic Date (65%) | Archaeomagnetic Date (95%) |
|---|---|---|---|---|---|---|---|
| **NP1** | 27/49 | 7/11 | 353.0 | −63.1 | 4.8 | 1160–1259 or 1367–1383 AD | 1131–1438 AD |
| **NP2** | 18/32 | 7/11 | 7.6 | −56.9 | 4.1 | 1272–1300 or 1448–1604 AD | 1255–1361 or 1388–1639 AD |
| **NP5** | 6/6 | n/a | 18.8 | −62.1 | 6.7 | 1725–1967 AD * | 1498–2000 AD * |
| **NGA1** | 6/17 | 2/10 | 12.5 | −63.9 | 2.6 | 1678–1839 AD | 1609–1914 AD |
| | 13/17 | 8/10 | 11.9 | −62.8 | 4.9 | 1592–1842 AD | 1441–1916 AD |

\* Site NP is known to pre-date the 1931 AD Napier earthquake.

### 3.2. Ngārara

A preliminary archaeomagnetic study of the Ngārara site is described by Robinson [29]. Seventeen specimens from ten stones from hāngī NGA1 were analysed by AFD. Demagnetization data for two specimens are shown in Figure 6f,g. Specimen NGA1-2-3A carries a single, stable component of TRM (Dec = 16.1°, Inc = −59.9°, MAD = 1.6°). NGA1-6-1A shows two components, the low coercivity component, removed at peak fields below 20 mT, is similar in direction to the TRM of NGA1-2-3A, but is not well-resolved (Dec = 6.7°, Inc = −74.8°, MAD = 9.0°). The fact that the higher coercivity component does not trend to the origin suggests that there is a third component carried in grains with still higher coercivity, which has not been isolated by AFD. ChRM data are shown and listed in Figure 7d and Appendix A. Two stones carry anomalous TRM directions that plot outside the range of Figure 7d and are not shown: these stones have clearly been disturbed and reoriented since cooling. Thirteen specimens from the remaining eight stones yield a cluster of data with Dec = 11.8°, Inc = −62.8°, $\alpha_{95}$ = 4.9°. However, the degree of scatter suggests that several of these stones have also moved slightly since cooling. Taking only the six specimens from stones NGA1-1 and NGA1-2, does not change the mean direction significantly, but produces a more satisfactory grouping and a higher degree of confidence in the mean: Dec = 12.5°, Inc = −63.9°, $\alpha_{95}$ = 2.6°.

Figure 8 shows the mean ChRM directions for each of the four features superimposed on the reference PSV curve, NZPSV1k.2023. Despite relatively large uncertainties, it is clear from the differences in declination that the three Napier features span several centuries, while the date of NGA1 is constrained between the mid-17th and mid-20th centuries.

### 3.3. Archaeomagnetic Dating

The principles and process of obtaining an archaeomagnetic date are illustrated for NGA1 in Figure 9, which is compiled from the output of the MATLAB archaeomagnetic dating tool of Pavón-Carrasco et al. [25]. The upper subplots show the NZPSV1k declination and inclination records in red. The ChRM declination and inclination are in black (relocated to 40° S, 175° E, using a virtual geomagnetic pole (VGP) transformation), and their 95% confidence limits ($\alpha_{95}$ for inclination and $\alpha_{95}$/cos(Inc) for declination). The probability density functions for fits to the NZPSV1k declination and inclination are shown in the middle pair of subplots, with 65% and 95% levels of confidence indicated, respectively. The lowermost subplot then gives the combined probably density function, with the results noted in the panel on the left. Declination and inclination distributions for NGA1 are mutually consistent, yielding an overall probability density function that has a single peak, and quasi-symmetrical shape. The median date of ca. 1740-1770 AD has a 65% confidence range of about ±80 years, or ±155 years for 95% confidence. This means the feature is likely to predate European colonisation in the Kapiti area and correspond to the period of the early-mid eighteenth century when different iwi were moving into and around the region.

This is somewhat younger than the mid-late seventeenth century radiocarbon date obtained from the nearby midden, NGA089, suggesting that they may not be contemporaneous. Alternatively, one or both features might have been used over protracted periods of time, with the radiocarbon date reflecting early use of the midden, while the archaeomagnetic date gives the last use of the hāngī.

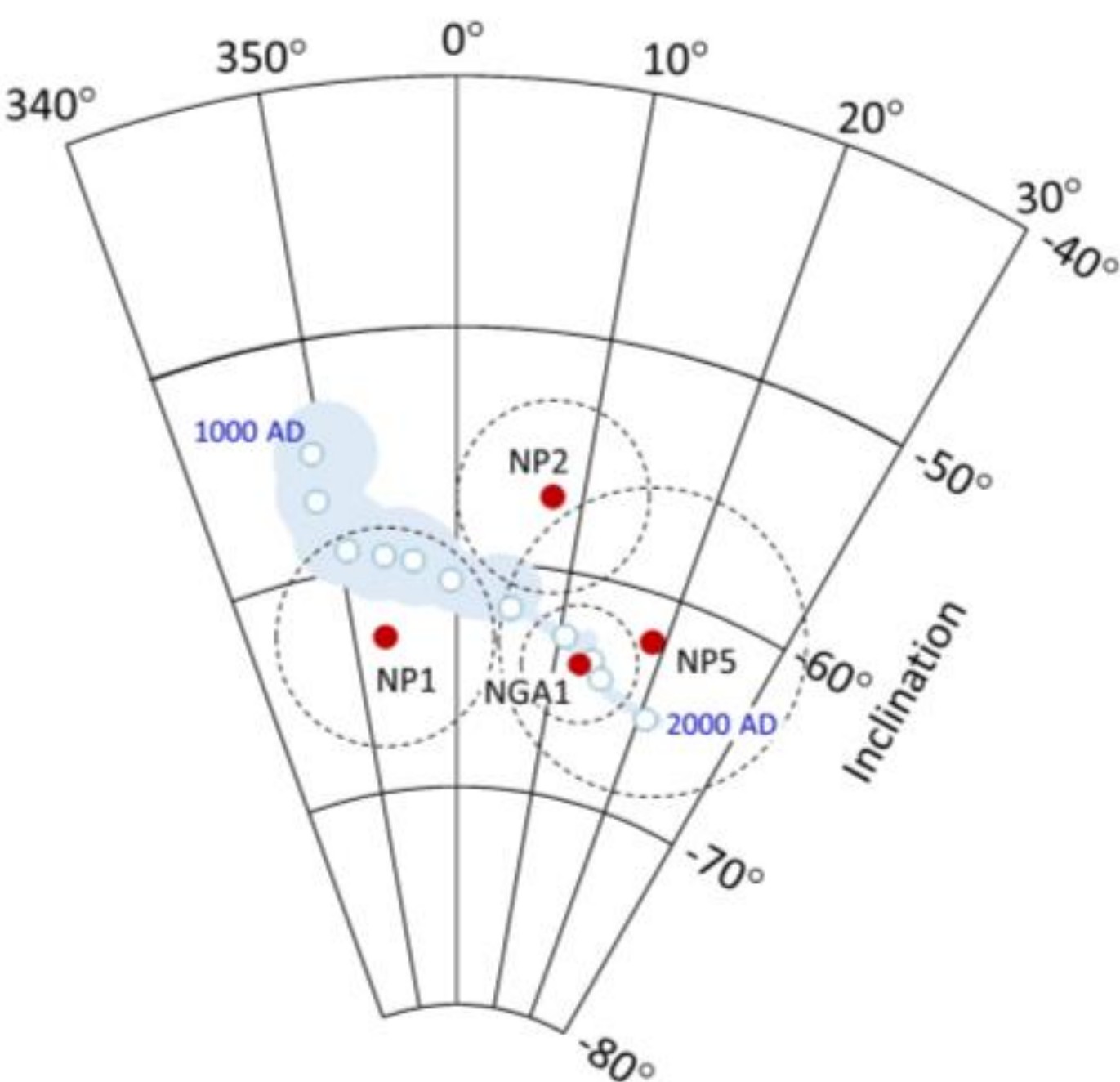

**Figure 8.** Partial stereographic (Wulff) projection showing the NZPSV1k.2023 reference curve at 100-year intervals (blue circles), and its 95% confidence (light blue shading). Mean ChRM directions and their $\alpha_{95}$s for NP1, NP2, NP5 and NGA1 are shown as red dots and black dashed circles, respectively.

A summary of dating results for the Napier site features is shown in Figure 10. The most likely dates are largely determined by the declination matches to NZPSV1k, with inclination fitting less well. NP1 has a westerly declination (353.0°), making it the oldest feature, most likely ca. 1210 ± 50 AD at 65% confidence (however, at 95% confidence, NP1 may be as late as 1438 AD). NP2 is somewhat younger, with a best estimated date of 1525 ± 75 AD (65%) and, at 95% confidence, between 1388 and 1639 AD. NP5 is the youngest dated feature at the Napier site, estimated at 1850 ± 120 AD (65%) or between 1500 and 2000 AD (95%).

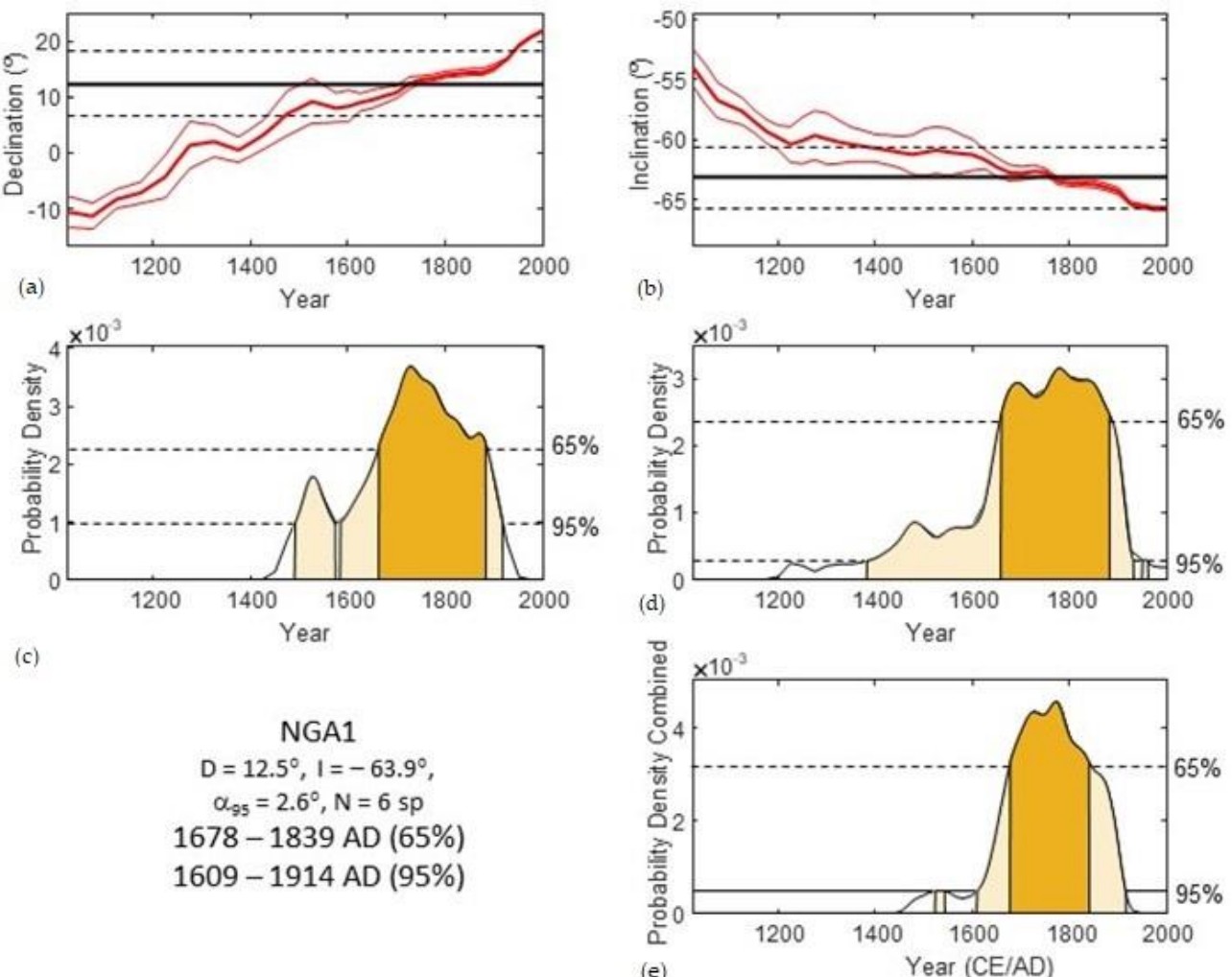

**Figure 9.** Archaeomagnetic date estimation for hāngī NGA1, using NZPSV1k.2023 as reference curve and the Matlab dating tool of Pavón-Carrasco et al. (2011) [25]. (**a**,**b**) Master PSVCs of declination and inclination (represented in red with bands of 95% confidence). The archaeomagnetic ChRM directions and their α95s are shown in black; (**c**,**d**) individual probability density functions for declination and inclination, with 65% and 95% confidence levels shown; (**e**) combined probability density function at 65% and 95% levels of confidence.

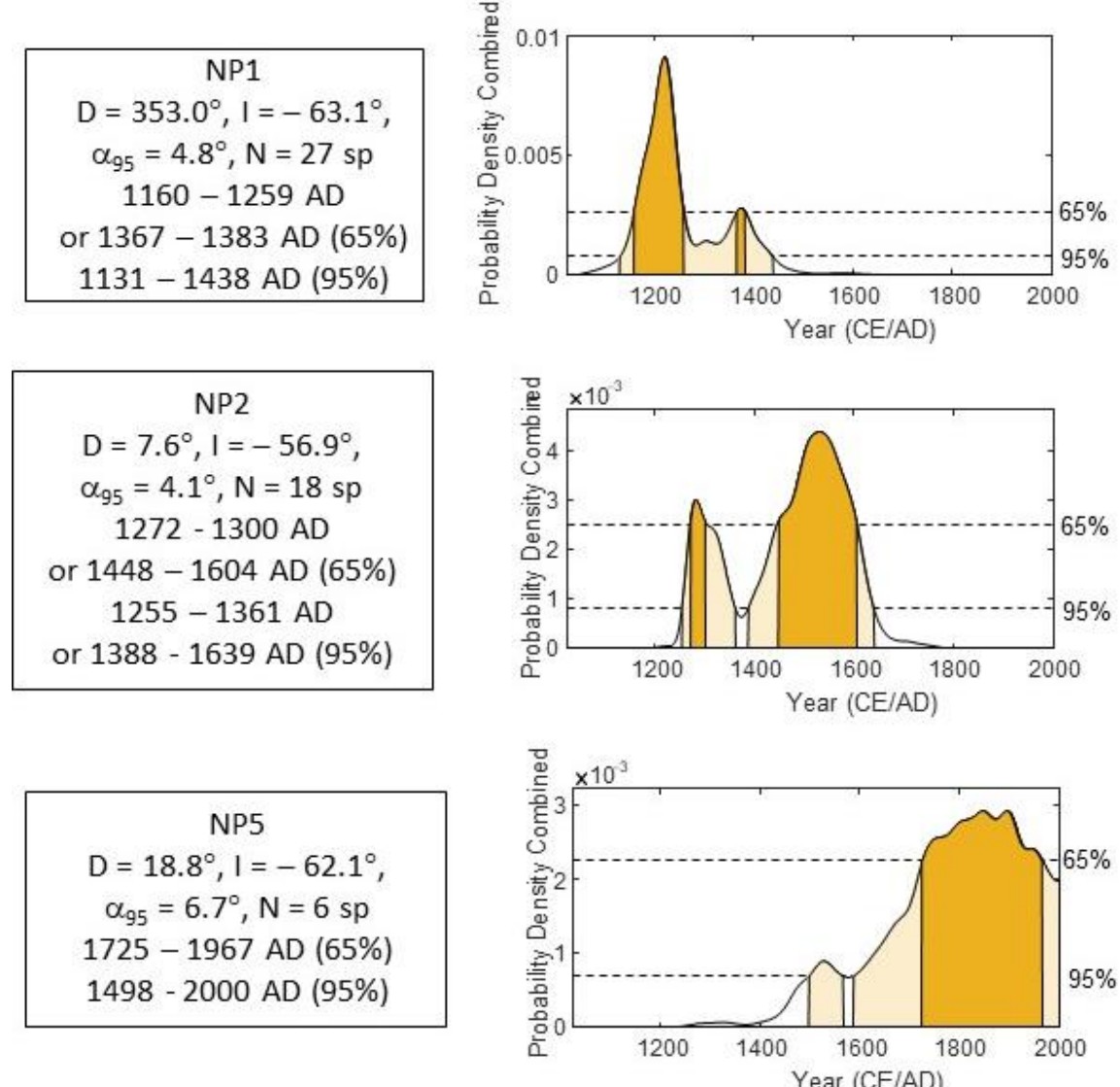

**Figure 10.** Archaeomagnetic date estimates for NP1, NP2 and NP5. Overall probability density functions and dates are estimated at 65% and 95% levels of confidence.

## 4. Discussion

The Māori early settlers of New Zealand left very little material that is suitable for archaeomagnetic study or dating. They did not make pottery or bricks or build kilns. Objective and/or absolute methods of dating their sites of occupation are also sparse. Radiocarbon dating of charcoal from fires or hāngī is often marred by inbuilt age [30], while the dating of shells, frequently found in both middens and hāngī, is complicated by uncertainty in the marine reservoir correction, ΔR [31]. The archaeomagnetic records carried by hāngī stones therefore provide a unique opportunity for dating prehistoric sites and for tracing the history of early occupation in New Zealand.

The thermoremanent magnetization imparted to hāngī stones during the heating and cooling process is typically strong and stable. It is often carried by magnetic minerals (magnetite, titanomagnetite and/or possibly maghaemite) that are themselves formed or enhanced in the heating process [12,13]. Further chemical alteration to the magnetic minerals is unlikely, due to the preservation of the archaeological features and their relative youth.

Hāngī stones are, however, not the ideal archaeomagnetic recorders, due to the likelihood that they may have been disturbed and reoriented after cooling. For this reason, it is usual to sample between ten and twelve stones from each feature, and to look for

clustered ChRM directions which may suggest the in situ orientation of the stones. The method of calculation of the best overall ChRM direction varies from feature to feature, depending on the nature of the data. In general, data may be disregarded at two stages. First, if the ChRM direction of an individual specimen is more than two circular standard deviations ($2\theta_{63}$) from the mean ChRM of its stone, then it is disregarded, and the stone mean recalculated. This process is repeated until all remaining specimen ChRM directions lie within $2\theta_{63}$ of the stone mean. Secondly, if the resulting mean ChRM direction of a stone is more than two circular standard deviations from the overall mean for the hāngī or feature, it is considered to lie outside the population of directions from undisturbed stones, and is disregarded as having suffered significant disturbance. The resulting hāngī–or feature mean ChRM is then considered to represent the record of undisturbed or minimally disturbed stones, and as such, as a record of the geomagnetic field at the time the stones last cooled to ambient temperature. In this study we calculate feature mean ChRM directions by averaging specimen data, rather than sample mean data. Poojary [12] has shown that the actual mean directions rarely differ significantly. However, the calculated confidence in the mean is inevitably poor if only a small number of sample means are averaged. We suggest that the larger number of specimens (whose data meet selection criteria) gives a better indication of the true confidence in their mean ChRM, particularly for the purpose of dating.

Obtaining a feature mean ChRM direction to a high degree of confidence is therefore a time-consuming process. Alternating field demagnetization is generally quicker than thermal demagnetization and usually yields satisfactory results. However, AFD demagnetizes progressively by grain coercivity, and so does not give a sequential profile of the magnetization process, as the stone cooled in the way that progressive THD, which demagnetizes grains according to their blocking temperatures does. If a stone has been disturbed part-way through the cooling process, it is therefore more likely that a ChRM recorded in the low blocking temperature part of the grain spectrum can be retrieved by THD rather than by AFD.

By contrast, the baked floor of an oven, hāngī or fireplace is unlikely to have been disturbed since its last use, so material can be sampled with confidence of its being in situ. Unfortunately, the material sampled from the Napier site was coarse grained and liable to grain movement within the sample box during sampling, handling and measurement. In future studies, it is recommended to lock the grains in place by infusing the material with a non-magnetic consolidating hardener before sampling, e.g. [32,33]. In addition, being derived from sand, the concentration of magnetic mineral and strength of magnetization are low compared with the hāngī stones, and the grain-size larger than the ideal single-domain size.

Application of the dating tool is straightforward if a good reference curve is available for the region, or if the site is situated in a region of the world for which a reliable geomagnetic model is available. For New Zealand, the period of human occupation is known to be less than 1000 years—short enough correlation with NZPSV1k.2023 yields unambiguous results. In regions of longer occupation, it may be necessary to apply independent constraints to the age or date of a feature, in order to avoid ambiguity or to distinguish between more than one match to the reference or modelled PSV curve.

## 5. Conclusions

We have presented two examples of the application of palaeomagnetic techniques in archaeological settings in New Zealand, and have deduced numerical ages for the features sampled, which was not possible by other means. In the case of the Napier/Ahuriri site on the east coast, we have presented detailed and extensive data from both thermal and alternating field demagnetization. We have shown that the area was occupied, possibly intermittently, for 400–500 years—from soon after the first Māori arrival in New Zealand, until European arrival, possibly longer. At Ngārara, on the west coast, archaeomagnetic data are preliminary, and ideally should be augmented by the results of thermal demagnetization

experiments, as well as rock magnetic investigations of the remanence carrier. Nonetheless, we have been able to place the site within the known history of the area—possibly only decades before the arrival of Europeans. Radiocarbon dating of a nearby midden suggests that this site has also seen prolonged occupation. Further archaeomagnetic work here should help unravel the complex history of the area.

**Author Contributions:** Conceptualization: S.P. and G.T.; methodology: S.P. and G.T.; formal analyses S.P., F.R. and G.T.; investigation: S.P., F.R. and G.T.; writing—original draft preparation: G.T.; writing—review and editing S.P., F.R. and G.T.; supervision: G.T. All authors have read and agreed to the published version of the manuscript.

**Funding:** This research received no external funding.

**Data Availability Statement:** A summary of the data produced in this research is included in Appendix A.

**Conflicts of Interest:** The authors declare no conflict of interest.

## Appendix A

Specimen level ChRM directions, determined by principal component analysis on selected data from progressive demagnetization procedures, stone mean and feature mean ChRM directions and statistics. Specimen and stone-level data in italics have been disregarded, since they lie outside two circular standard deviations of the appropriate stone or feature mean direction. Dec = declination, Inc = inclination, MAD = maximum angular deviation (from principal component analysis), $\alpha_{95}$ = semiangle of the cone of 95% confidence in the mean direction; $\theta_{63}$ = circular standard deviation of the set of directions.

| NGA1 | | | | | | | |
|---|---|---|---|---|---|---|---|
| Specimen ID | Demagnetization Procedure | ChRM Range | n (Demag Steps) | Dec (°) | Inc (°) | MAD (°) | Angle between Specimen and Overall (13 Specimen) Mean ChRM (°) |
| NGA1-1-1A | AFD (2.5–80 mT) | 20–70 mT | 7 | 15.1 | −62.2 | 1.7 | 1.6 |
| NGA1-1-1B | AFD (2.5–80 mT) | 2.5–70 mT | 11 | 10.7 | −63.3 | 1.7 | 0.74 |
| NGA1-1-1C | AFD (2.5–80 mT) | 2.5–70 mT | 11 | 9.6 | −66.3 | 1.3 | 3.64 |
| NGA1-2-1A | AFD (2.5–80 mT) | 2.5–80 mT | 12 | 17.4 | −66.2 | 1.3 | 4.14 |
| NGA1-2-3A | AFD (2.5–80 mT) | 15–80 mT | 9 | 16.1 | −59.9 | 1.6 | 3.53 |
| NGA1-2-4A | AFD (2.5–80 mT) | 15–80 mT | 9 | 5.6 | −65.0 | 1.2 | 3.54 |
| *NGA1-4-1B* | AFD (2.5–80 mT) | *2.5–60 mT* | 10 | *191.6* | *−82.8* | *1.6* | *34.4* |
| *NGA1-4-2B* | AFD (2.5–80 mT) | *5–70 mT* | 10 | *41.9* | *−81.6* | *1.4* | *20.3* |
| *NGA1-4-3B* | AFD (2.5–80 mT) | *2.5–60 mT* | 10 | *183.1* | *−85.3* | *1.9* | *31.85* |
| NGA1-5-2A | AFD (2.5–80 mT) | 15–80 mT | 9 | 10.3 | −53.0 | 1.3 | 9.84 |
| NGA1-6-1A | AFD (2.5–80 mT) | 0–20 mT | 6 | 6.7 | −74.8 | 9.0 | 12.14 |
| NGA1-7-2B | AFD (2.5–90 mT) | 15–90 mT | 10 | 31.2 | −58.6 | 2.4 | 10.29 |
| NGA1-8-2A | AFD (2.5–80 mT) | 25–80 mT | 7 | 335.8 | −67.9 | 1.1 | 15.63 |
| NGA1-8A-1B | AFD (2.5–80 mT) | 5–70 mT | 10 | 353.5 | −66.8 | 1.1 | 8.85 |
| *NGA1-9-1A* | AFD (2.5–90 mT) | *40–90 mT* | 6 | *211.7* | *−67.8* | *4.5* | *48.62* |
| NGA1-10-1A | AFD (2.5–80 mT) | 30–80 mT | 6 | 30.8 | −50.0 | 2.1 | 16.4 |
| NGA1-11-2A | AFD (2.5–90 mT) | 25–80 mT | 7 | 10.0 | −53.6 | 2.6 | 9.25 |

**NGA1 Mean ChRM: Dec = 11.9°, Inc = −62.8°, N = 13 specimens from 8 stones, $\alpha_{95}$ = 4.9°, $2\theta_{63}$ = 19.0°**

**NGA1 Mean ChRM: Dec = 12.5°, Inc = −63.9°, N = 6 specimens from 2 stones, $\alpha_{95}$ = 2.6°, $2\theta_{63}$ = 6.84°**

| NP1 | | | | | | | |
|---|---|---|---|---|---|---|---|
| Specimen ID | Demagnetization Procedure | ChRM Range | n (Demag Steps) | Dec (°) | Inc (°) | MAD (°) | Angle between Specimen and Stone Mean ChRM (°) |
| NP1-1-A1 | AFD (2.5–99 mT) THD (300–600 °C) | 2.5–99 mT 300-600 °C | 25 | 314.2 | −43.3 | 0.9 | 5.3 |
| NP1-1-A2 | THD-P (100–600 °C) | 100–600 °C | 18 | 315.0 | −44.9 | 1.0 | 4.8 |

| | | | | | | | |
|---|---|---|---|---|---|---|---|
| NP1-1-B1 | THD (50–640 °C) | 50–640 °C | 19 | 297.4 | −45.5 | 0.8 | 8.0 |
| NP1-1-B2 | THD-P (100–600 °C) | 200–600 °C | 17 | 304.0 | −56.4 | 1.8 | 9.8 |
| NP1-1-C1 | THD (50–620 °C) | 50–620 °C | 28 | 298.8 | −45.9 | 0.8 | 7.0 |
| NP1-1-D1 | THD (50–620 °C) | 50–620 °C | 28 | 312.6 | −46.1 | 0.8 | 2.8 |
| NP1-1-E1 | AFD (2.5–99 mT) THD (300–600 °C) | 10–99 mT 300–600 °C | 28 | 318.2 | −49.5 | 0.7 | 6.7 |
| NP1-1-F1 | THD (50–640 °C) | 50–640 °C | 19 | 307.6 | −47.1 | 0.8 | 0.8 |
| NP1-1-H1 | THD (50–620 °C) | 50–620 °C | 28 | 308.2 | −44.7 | 0.9 | 2.3 |
| NP1-1-H2 | AFD (2.5–99 mT) THD (300–600 °C) | 3.0–99 mT 300–600 °C | 32 | 311.0 | −45.2 | 0.7 | 2.4 |

*Mean ChRM, stone NP1-1: Dec = 308.8°, Inc = −47.0°, N = 10 specimens, $\alpha_{95}$ = 3.6°, $2\theta_{63}$ = 12.0°; angle between stone mean and overall NP1 mean ChRM = 29.14°*

| | | | | | | | |
|---|---|---|---|---|---|---|---|
| NP1-2-A1 | AFD (2.5–99 mT) THD (300–600 °C) | 18–99 mT 300—600 °C | 27 | 325.3 | −65.0 | 1.1 | 2.9 |
| NP1-2-B1 | THD (50–640 °C) | 100–640 °C | 18 | 326.7 | −65.1 | 1.1 | 2.3 |
| *NP1-2-C1* | *THD (50–620 °C)* | *120–620 °C* | *25* | *344.0* | *−47.2* | *1.2* | *18.7* |
| NP1-2-D1 | THD (50–640 °C) | 50–640 °C | 19 | 337.4 | −64.9 | 1.1 | 2.3 |
| NP1-2-E1 | THD-P (100-600 °C) | 100–600 °C | 18 | 347.6 | −64.4 | 1.1 | 6.6 |
| NP1-2-E1 | AFD (2.5–99 mT) THD (300–600 °C) | 10–99 mT 300–600 °C | 26 | 323.7 | −63.4 | 0.9 | 3.9 |

**Mean ChRM, stone NP1-2: Dec = 332.1°, Inc = −64.8°, N = 5 specimens, $\alpha_{95}$ = 4.2°, $2\theta_{63}$ = 8.8°; angle between stone mean and overall NP1 mean ChRM = 9.29°**

| | | | | | | | |
|---|---|---|---|---|---|---|---|
| NP1-3-A1 | AFD (2.5–99 mT) THD (300–600 °C) | 18–99 mT 300–600 °C | 27 | 11.0 | −79.2 | 0.8 | 4.2 |
| NP1-3-B1 | THD (50–660 °C) | 50–640 °C | 30 | 28.1 | −77.0 | 1.1 | 0.2 |
| NP1-3-C1 | THD (50–640 °C) | 50–640 °C | 19 | 30.2 | −75.6 | 1.1 | 1.3 |
| NP1-3-C2 | THD-P (100–600 °C) | 150–600 °C | 17 | 35.7 | −75.0 | 1.1 | 2.6 |
| NP1-3-E1 | AFD (2.5–99 mT) THD (300–600 °C) | 10–99 mT 300–600 °C | 28 | 31.4 | −76.4 | 0.7 | 0.9 |

**Mean ChRM, stone NP1-3: Dec = 28.2°, Inc = −76.8°, N = 5 specimens, $\alpha_{95}$ = 2.5°, $2\theta_{63}$ = 5.2°; angle between stone mean and overall NP1 mean ChRM = 17.70°**

| | | | | | | | |
|---|---|---|---|---|---|---|---|
| *NP1-4-B1* | *THD (50–640 °C)* | *50–640 °C* | *30* | *209.4* | *−71.6* | *1.6* | *19.5* |
| NP1-4-C1 | AFD (2.5–99 mT) THD (300–600 °C) | 30–99 mT 300-600 °C | 24 | 278.5 | −71.0 | 0.6 | 1.7 |
| NP1-4-D1 | THD (50–640 °C) | 50–640 °C | 19 | 274.3 | −70.1 | 0.7 | 1.1 |
| NP1-4-E1 | THD-P (100–600 °C) | 100–600 °C | 18 | 266.9 | −72.4 | 0.8 | 2.4 |

*Mean ChRM, stone NP1-4: Dec = 273.4°, Inc = −71.2°, N = 3 specimens, $\alpha_{95}$ = 3.3°, $2\theta_{63}$ = 4.4°; angle between stone mean and overall NP1 mean ChRM = 29.48°*

| | | | | | | | |
|---|---|---|---|---|---|---|---|
| *NP1-5-A1* | *AFD (2.5–99 mT) THD (300–600 °C)* | *2.5–99 mT 300-600 °C* | *26* | *332.9* | *−31.4* | *1.0* | *33.5* |
| NP1-5-A2 | THD-P (100–625 °C) | 100–325 °C | 9 | 18.6 | −68.4 | 5.6 | 12.5 |
| NP1-5-B1 | THD (50–620 °C) | 50–460 °C | 17 | 332.6 | −57.5 | 2.6 | 10.6 |
| NP1-5-C1 | THD (50–640 °C) | 50–420 °C | 8 | 351.4 | −58.9 | 5.5 | 3.9 |

**Mean ChRM, stone NP1-5: Dec = 351.2°, Inc = −62.8°, N = 3 specimens, $\alpha_{95}$ = 18.3°, $2\theta_{63}$ = 23.8°; angle between stone mean and overall NP1 mean ChRM = 0.87°**

| | | | | | | | |
|---|---|---|---|---|---|---|---|
| NP1-6-A1 | AFD (2.5–99 mT) THD (300–600 °C) | 2.5–99 mT 300–600 °C | 34 | 10.6 | −51.5 | 0.9 | 3.3 |
| NP1-6-A2 | THD (50–660 °C) | 50–580 °C | 27 | 7.9 | −50.5 | 1.0 | 3.0 |
| NP1-6-B1 | THD (50–640 °C) | R.T.-460 °C | 11 | 4.0 | −56.6 | 7.5 | 3.5 |
| NP1-6-B2 | THD-P (100–625 °C) | R.T.-400 °C | 11 | 1.1 | −54.4 | 8.8 | 3.1 |

**Mean ChRM, stone NP1-6: Dec = 6.0°, Inc = −53.3°, N = 4 specimens, $\alpha_{95}$ = 4.3°, $2\theta_{63}$ = 7.5°; angle between stone mean and overall NP1 mean ChRM = 11.91°**

| NP1-7-A1 | THD (50–640 °C) | 50–640 °C | 19 | 357.9 | −47.4 | 0.8 | 5.9 |
|---|---|---|---|---|---|---|---|
| NP1-7-B1 | THD (50–640 °C) | 50–420 °C | 19 | 354.0 | −47.7 | 2.8 | 6.3 |
| NP1-7-C1 | THD-P (100-600 °C) | 100–500 °C | 18 | 17.1 | −56.1 | 2.9 | 11.1 |
| NP1-7-C2 | THD (50–620 °C) | 50–320 °C | 14 | 344.8 | −60.1 | 3.9 | 10.1 |

**Mean ChRM, stone NP1-7: Dec = 358.5°, Inc = −53.3°, N = 4 specimens, $\alpha_{95}$ = 11.4°, $2\theta_{63}$ = 19.9°; angle between stone mean and overall NP1 mean ChRM = 10.21°**

| NP1-8-A1 | AFD (2.5–99 mT) THD (300–600 °C) | 15–99 mT 300–600 °C | 28 | 50.2 | −46.2 | 1.4 | 0.8 |
|---|---|---|---|---|---|---|---|
| NP1-8-B1 | THD (50–620 °C) | 75–620 °C | 28 | 52.2 | −44.1 | 2.0 | 2.1 |
| NP1-8-B2 | THD-P (100–600 °C) | 150–600 °C | 18 | 48.0 | −45.8 | 2.2 | 1.6 |

*Mean ChRM, stone NP1-8: Dec = 50.2°, Inc = −45.4°, N = 3 specimens, $\alpha_{95}$ = 2.8°, $2\theta_{63}$ = 3.7°; angle between stone mean and overall NP1 mean ChRM = 36.19°*

| *NP1-9-A1* | *THD (50–640 °C)* | *50–175 °C* | *3* | *306.3* | *−60.4* | *10.6* | *24.6* |
|---|---|---|---|---|---|---|---|
| NP1-9-B1 | THD (50–620 °C) | 50–180 °C | 7 | 1.2 | −79.9 | 2.4 | 4.8 |
| NP1-9-C1 | THD-P (100–600 °C) | R.T.-225 °C | 5 | 4.6 | −70.2 | 4.7 | 4.9 |

**Mean ChRM, stone NP1-9: Dec = 3.4°, Inc = −75.1°, N = 2 specimens, $\alpha_{95}$ = 21.4°, $2\theta_{63}$ = 13.8°; angle between stone mean and overall NP1 mean ChRM = 12.52°**

| NP1-10-A1 | THD (50–640 °C) | 50–640 °C | 19 | 333.4 | −49.4 | 1.4 | 2.4 |
|---|---|---|---|---|---|---|---|
| NP1-10-B1 | AFD (2.5–99 mT) THD (300–600 °C) | 2.5–99 mT 300–600 °C | 34 | 340.2 | −50.6 | 0.8 | 2.3 |
| NP1-10-D1 | THD-P (100–600 °C) | 100–600 °C | 18 | 337.6 | −51.6 | 1.6 | 1.1 |
| NP1-10-E1 | THD (50–625 °C) | 50–625 °C | 28 | 334.8 | −50.9 | 0.7 | 1.1 |

**Mean ChRM, stone NP1-10: Dec = 336.5°, Inc = −50.7°, N = 4 specimens, $\alpha_{95}$ = 2.4°, $2\theta_{63}$ = 4.3°; angle between stone mean and overall NP1 mean ChRM = 15.23°**

| NP1-11-A1 | THD-P (100–600 °C) | 100-500 °C | 14 | 68.5 | 8.6 | 1.2 | 6.7 |
|---|---|---|---|---|---|---|---|
| NP1-11-B1 | THD (50–640 °C) | 50–480 °C | 22 | 55.1 | 6.8 | 2.9 | 6.7 |

*Mean ChRM, stone NP1-11: Dec = 61.8°, Inc = 7.8°, N = 2 specimens, $\alpha_{95}$ = 29.6°, $2\theta_{63}$ = 18.3°; angle between stone mean and overall NP1 mean ChRM = 87.65°*

**NP1 Mean ChRM: Dec = 353.0°, Inc = −63.1°, N = 27 specimens from 7 stones, $\alpha_{95}$ = 4.8°, $2\theta_{63}$ = 28.8°**

**NP2**

| Specimen ID | Demagnetization Procedure | ChRM Range | n (Demag Steps) | Dec (°) | Inc (°) | MAD (°) | Angle between Specimen and Stone Mean ChRM (°) |
|---|---|---|---|---|---|---|---|
| NP2-1-A1 | THD (50–640 °C) | 50–640 °C | 19 | 5.3 | −57.7 | 0.5 | 1.5 |
| NP2-1-B1 | THD (50–660 °C) | 50–660 °C | 31 | 7.2 | −60.2 | 1.0 | 1.7 |
| NP2-1-B2 | THD-P (100–625 °C) | 100–625 °C | 19 | 10.1 | −59.8 | 1.0 | 1.8 |
| NP2-1-C1 | AFD (2.5–99 mT) | 2.5–99 mT | 24 | 6.2 | −56.3 | 1.0 | 2.3 |
| NP2-1-D1 | THD (50–660 °C) | 50–660 °C | 31 | 9.4 | −58.6 | 1.8 | 0.9 |
| *NP2-1-E1* | *AFD (2.5–99 mT)* | *10–99 mT* | *18* | *345.9* | *−62.4* | *0.7* | *11.3* |

**Mean ChRM, stone NP2-1: Dec = 7.6°, Inc = −58.5°, N = 5 specimens, $\alpha95$ = 1.8°, $2\theta_{63}$ = 3.8°; angle between stone mean and overall NP2 mean ChRM = 1.6°**

| *NP2-2-A1* | *AFD (2.5–99 mT)* | *2.5–15 mT* | *4* | *339.2* | *−35.1* | *2.0* | *20.7* |
|---|---|---|---|---|---|---|---|
| *NP2-2-A2* | *THD-P (100–625 °C)* | *100–400 °C* | *10* | *311.7* | *−32.0* | *7.9* | *36.7* |
| NP2-2-B1 | THD (50–620 °C) | 50–400 °C | 15 | 352.1 | −54.5 | 4.6 | 1.7 |
| NP2-2-C1 | THD (50–640 °C) | 50–400 °C | 19 | 355.4 | −51.7 | 10.2 | 1.7 |

**Mean ChRM, stone NP2-2: Dec = 353.8°, Inc = −53.1°, N = 2 specimens, $\alpha_{95}$ = 7.5°, $2\theta_{63}$ = 4.9°; angle between stone mean and overall NP2 mean ChRM = 8.76°**

| NP2-3-B1 | THD (50–620 °C) | 50–320 °C | 14 | 5.6 | −58.2 | 5.6 | 0.4 |
|---|---|---|---|---|---|---|---|
| NP2-3-B2 | THD-P (100–600 °C) | 100–325 °C | 7 | 4.4 | −57.6 | 3.7 | 0.4 |

**Mean ChRM, stone NP2-3: Dec = 5.0°, Inc = −57.9°, N = 2 specimens, $\alpha_{95}$ = 1.9°, $2\theta_{63}$ = 1.2°; angle between stone mean and overall NP2 mean ChRM = 1.72°**

| | | | | | | | |
|---|---|---|---|---|---|---|---|
| NP2-4-A1 | THD (50–620 °C) | 160–460 °C | 14 | 1.2 | −50.3 | 3.2 | 2.2 |
| NP2-4-B1 | THD-P (100–600 °C) | 100–375 °C | 9 | 1.2 | −45.8 | 3.3 | 2.3 |

**Mean ChRM, stone NP2-4: Dec = 1.2°, Inc = −48.1°, N = 2 specimens, $\alpha_{95}$ = 9.8°, $2\theta_{63}$ = 6.4°; angle between stone mean and overall NP2 mean ChRM = 9.61°**

| | | | | | | | |
|---|---|---|---|---|---|---|---|
| NP2-5-A1 | THD (50–620 °C) | 50–520 °C | 20 | 117.8 | −79.1 | 2.1 | 1.2 |
| NP2-5-B1 | THD-P (100–600 °C) | 100–425 °C | 11 | 105.4 | −79.3 | 5.3 | 1.2 |

*Mean ChRM, stone NP2-5: Dec = 111.7°, Inc = −79.3°, N = 2 specimens, $\alpha_{95}$ = 5.1°, $2\theta_{63}$ = 3.3°; angle between stone mean and overall NP2 mean ChRM = 37.02°*

| | | | | | | | |
|---|---|---|---|---|---|---|---|
| NP2-6-A1 | THD (50–620 °C) | 50–200 °C | 8 | 47.0 | −20.4 | 5.0 | 1.1 |
| NP2-6-B1 | THD-P (100–600 °C) | 100–225 °C | 4 | 45.6 | −22.1 | 3.6 | 1.0 |

*Mean ChRM, stone NP2-6: Dec = 46.3°, Inc = −21.3°, N = 2 specimens, $\alpha_{95}$ = 4.7°, $2\theta_{63}$ = 3.0°; angle between stone mean and overall NP2 mean ChRM = 45.46°*

| | | | | | | | |
|---|---|---|---|---|---|---|---|
| *NP2-7-A1* | *THD-P (100–625 °C)* | *150–375 °C* | *8* | *323.9* | *−46.5* | *4.4* | *39.1* |
| NP2-7-B1 | THD (50–640 °C) | 50–500 °C | 12 | 26.8 | −40.8 | 1.2 | 6.1 |
| NP2-7-C1 | THD (50–620 °C) | 50–540 °C | 24 | 15.2 | −50.0 | 1.3 | 6.2 |

**Mean ChRM, stone NP2-7: Dec = 21.5°, Inc = −45.5°, N = 2 specimens, $\alpha_{95}$ = 27.0°, $2\theta_{63}$ = 17.3°; angle between stone mean and overall NP2 mean ChRM = 14.29°**

| | | | | | | | |
|---|---|---|---|---|---|---|---|
| NP2-8-A1 | THD-P (100–600 °C) | R.T.-300 °C | 6 | 24.0 | −56.5 | 4.0 | 3.3 |
| NP2-8-B1 | THD (50–620 °C) | 50–320 °C | 14 | 12.6 | −54.8 | 10.3 | 3.3 |

**Mean ChRM, stone NP2-8: Dec = 18.2°, Inc = −55.8°, N = 2 specimens, $\alpha_{95}$ = 14.5°, $2\theta_{63}$ = 9.4°; angle between stone mean and overall NP2 mean ChRM = 5.97°**

| | | | | | | | |
|---|---|---|---|---|---|---|---|
| NP2-9-A1 | THD (50–650 °C) | 50–240 °C | 10 | 13.6 | −73.2 | 8.8 | 5.4 |
| NP2-9-B1 | THD-P (100-600 °C) | R.T.-250 °C | 6 | 352.9 | −73.1 | 4.1 | 5.9 |
| NP2-9-C1 | THD (50–650 °C) | 50–260 °C | 11 | 6.5 | −58.9 | 9.3 | 9.7 |

**Mean ChRM, stone NP2-9: Dec = 4.8°, Inc = −68.6°, N = 3 specimens, $\alpha_{95}$ = 13.6°, $2\theta_{63}$ = 17.8°; angle between stone mean and overall NP2 mean ChRM = 11.77°**

| | | | | | | | |
|---|---|---|---|---|---|---|---|
| NP2-10-A1 | THD (50–650 °C) | 50–220 °C | 9 | 54.3 | −48.1 | 5.3 | 1.3 |
| NP2-10-B1 | THD (50–640 °C) | 50–350 °C | 6 | 51.7 | −50.0 | 5.3 | 1.2 |
| *NP2-10-D1* | *THD-P (100–600 °C)* | *100–250 °C* | *5* | *89.2* | *−36.4* | *8.0* | *29.1* |

*Mean ChRM, stone NP2-10: Dec = 53.0°, Inc = −49.1°, N = 2 specimens, $\alpha_{95}$ = 5.6°, $2\theta_{63}$ = 36.1°; angle between stone mean and overall NP2 mean ChRM = 27.84°*

| | | | | | | | |
|---|---|---|---|---|---|---|---|
| NP2-11-A1 | THD (50–640 °C) | 50–640 °C | 19 | 5.8 | 34.4 | 1.1 | 12.3 |
| NP2-11-B1 | THD-P (100–600 °C) | 100–600 °C | 18 | 21.0 | 47.6 | 1.9 | 7.8 |
| NP2-11-B2 | THD (50–650 °C) | 50–650 °C | 29 | 9.5 | 33.0 | 0.7 | 10.7 |
| NP2-11-C1 | THD (50–650 °C) | 50–575 °C | 26 | 45.4 | 40.6 | 1.8 | 19.6 |

*Mean ChRM, stone NP2-11: Dec = 19.7°, Inc = 39.9°, N = 4 specimens, $\alpha_{95}$ = 17.7°, $2\theta_{63}$ = 30.6°; angle between stone mean and overall NP2 mean ChRM = 97.34°*

**NP2 Mean ChRM: Dec = 7.6°, Inc = -56.9°, N = 18 specimens from 7 stones, $\alpha_{95}$ = 4.1°, $2\theta_{63}$ = 20.2°**

**NP5**

| Specimen ID | Demagnetization Procedure | ChRM Range | n (Demag Steps) | Dec (°) | Inc (°) | MAD (°) | Angle between Specimen and Stone Mean ChRM (°) |
|---|---|---|---|---|---|---|---|
| NP5-2 | AFD (2.5–90 mT) | 2.5–20 | 19 | 13.5 | −61.7 | 2.2 | 2.5 |
| NP5-3 | AFD (2.5–90 mT) | 2.5–12.5 | 5 | 12.1 | −50.7 | 3.1 | 12.0 |
| NP5-4 | AFD (2.5–90 mT) | 2.5–15 | 6 | 23.6 | −65.7 | 4.5 | 4.2 |
| NP5-6 | AFD (2.5–90 mT) | 5.0–25 | 9 | 33.2 | −67.2 | 4.2 | 8.0 |
| NP5-7 | AFD (2.5–90 mT) | 2.5–15 | 6 | 29.7 | −68.2 | 2.7 | 7.6 |
| NP5-8 | AFD (2.5–90 mT) | 5.0–20 | 17 | 9.6 | −57.6 | 4.3 | 6.4 |

**Mean ChRM, feature NP5: Dec = 18.8°, Inc = −62.1°, N = 6 specimens, $\alpha_{95}$ = 6.7°, $2\theta_{63}$ = 16.2°**

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
