# Peer review of "Using Palaeomagnetic Techniques to Date Indigenous Archaeological Sites in New Zealand"

_heritage, doi:10.3390/heritage6100345_

Round 1
Reviewer 1 Report
Dear Authors and Editor,
Poojary et al. present two case studies of the application of paleomagnetic dating to date indigenous Maori hangi features and thereby provide constrains for archeological research on the time and duration of the Maori occupation at these sites. The work shows well the promise of using this method for dating without still promising too much and therefore giving a realistic image of the possibilities and challenges of this method. The work is significant, logically organized, very well written, suitable for the journal and to me acceptable with minor revision. The suggested changes and corrections should be fairly fast applicaple and mainly clarify few points of the manuscript. Attached is also a pdf with comments in more detail and I hope the authors will find these helpful.
With warm regards,
Reviewer
The few main clarifications are listed below:
Add somewhere a sentence or two about using specimen means instead of sample means. In her MSc thesis Poojary showed that these are very similar within their error, and one could refer to her work there, but also it would be good to add in the paper for the paleomagnetic audience in a more clear way. See comment 4) for lines 311-313.
Lines 127-133. Has this been published elsewhere already? If yes, please add a reference, and if not, maybe it could be added to this manuscript. One figure would suffice with temperature vs time data in one sub figure and the consequent directional data in other sub figure.
Lines 304-307. Please add a sentence or two about the reasons for disregarding specimens from one individually oriented rock. What can be the reasonf for specimen disregarding, wouldn't that mean the sample (individually oriented rock) has also been disturbed and moved? Or is this related to uneven heating, i.e. some part of the rock does acquire correct direction, but some faster cooling parts do not? Or is it because some other heterogeneities in the rock?
Lines 311-313. After this would be a good location to add few sentences about whether one should use sample means or specimen means, and add a reference to Poojary's MSc thesis where these are shown to coincide very well for the NP features in question.
Figures and tables.
For Figure 1, a sketch of the past Maori hangi would be preferred. That should not be hard nor take too long time in some drawing program such as Inkscape for example.
Figure 3. Especially part (a) is not readable in current size (if the size is what I received in the Figures pdf). Either try to find a better resolution image or try zooming in closer.
Figure 7d. What is the difference between the two means shown?
Table 2. Remove extra text above Table caption. Also, please add the n/N, since it would be very interesting in terms of having an idea of the amount of work put in vs the amount of measurements that were determined reliable.

Author Response
USING PALAEOMAGNETIC TECHNIQUES TO DATE INDIGENOUS ARCHAEOLOGICAL SITES IN NEW ZEALAND
Shefali Poojary 1, Fergus Robinson 2 and Gillian Turner 3,*
Responses to reviewers’ comments (In italics. Line numbers refer to the version of the revised manuscript with all changes tracked). We have also added responses to the comments and suggestions the reviewers have made on annotated (pdf) copies of the manuscript – attached.
Reviewer 1
Poojary et al. present two case studies of the application of paleomagnetic dating to date indigenous Maori hangi features and thereby provide constrains for archeological research on the time and duration of the Maori occupation at these sites. The work shows well the promise of using this method for dating without still promising too much and therefore giving a realistic image of the possibilities and challenges of this method. The work is significant, logically organized, very well written, suitable for the journal and to me acceptable with minor revision. The suggested changes and corrections should be fairly fast applicaple and mainly clarify few points of the manuscript. Attached is also a pdf with comments in more detail and I hope the authors will find these helpful.
With warm regards,
Reviewer
The few main clarifications are listed below:
Add somewhere a sentence or two about using specimen means instead of sample means. In her MSc thesis Poojary showed that these are very similar within their error, and one could refer to her work there, but also it would be good to add in the paper for the paleomagnetic audience in a more clear way. See comment 4) for lines 311-313.
Explanatory sentence added (lines 336-341).
Lines 127-133. Has this been published elsewhere already? If yes, please add a reference, and if not, maybe it could be added to this manuscript. One figure would suffice with temperature vs time data in one sub figure and the consequent directional data in other sub figure.
Reference added (NZAA conference, 2016). The experiment has also been cited in Turner et al., 2020.
Lines 304-307. Please add a sentence or two about the reasons for disregarding specimens from one individually oriented rock. What can be the reasonf for specimen disregarding, wouldn't that mean the sample (individually oriented rock) has also been disturbed and moved? Or is this related to uneven heating, i.e. some part of the rock does acquire correct direction, but some faster cooling parts do not? Or is it because some other heterogeneities in the rock?
Possible reasons for this have been added (lines 230-232)
Lines 311-313. After this would be a good location to add few sentences about whether one should use sample means or specimen means, and add a reference to Poojary's MSc thesis where these are shown to coincide very well for the NP features in question.
Done: Explanatory sentence added (lines 336-341).
Figures and tables.
For Figure 1, a sketch of the past Maori hangi would be preferred. That should not be hard nor take too long time in some drawing program such as Inkscape for example.
Done – see new Figure 1.
Figure 3. Especially part (a) is not readable in current size (if the size is what I received in the Figures pdf). Either try to find a better resolution image or try zooming in closer.
We now use the highest resolutioin image available (Figure 3, high resolution separate file). This is a historic document, kindly ptovided to us by the museum in Napier. As such it has natural limitations. We have also expanded the figure caption to draw attention to the relevant features and particularly to differences between this 1865 map and the contemporary map, differences that resulted from the 1931 Napier earthquake.
Figure 7d. What is the difference between the two means shown?
This is explained in the text (lines 264 and 268), the note beneath the figure (N=13 and N=6 specimens)
Table 2. Remove extra text above Table caption. Also, please add the n/N, since it would be very interesting in terms of having an idea of the amount of work put in vs the amount of measurements that were determined reliable.
I‘m not sure what you mean here. The table has only a one-line descriptive caption as is conventional. We have added N/N0 for specimens and n/n0 for stones as requested.

Reviewer 2 Report
I welcomed the opportunity to review the paper 'Using Paleomagnetic techniques to date indigenous archaeological sites in New Zealand' by Poojary et al.
Overall the manuscript is well organized and written and after a set of comments are addressed I think that it is ready to be published.
Before that stage, I would like to view the cited figures which are not appended in the main manuscript and this is a major concern.
I suggest to merge figures 1 with 5 (deleting figure 4) I also suggest to merge figures 2 and 3 as one location map, increasing the font sizes and adding the Geographic north and scales for reference. In Figure 6, please add stereonets and Magnetization/ Temperature or mT to discuss the demagnetization trend, and stereonet to discuss if any anisotropy exists and/ or parasitic field is acquired during demagnetization. Please add uncertainties to all data points in figure 7 and mark ones which are not deemed reliable. Please merge figures 9 and 10. After all these changes, please update the manuscript accordingly.All other major and minor comments regarding manuscript organization, reference citations and discussion of the data and the results can be found in the attached PDF version of the manuscript.
Best wishes

The manuscript is well written. Few comments regarding the style and wording can be found within the annotated PDF.
Author Response
USING PALAEOMAGNETIC TECHNIQUES TO DATE INDIGENOUS ARCHAEOLOGICAL SITES IN NEW ZEALAND
Shefali Poojary 1, Fergus Robinson 2 and Gillian Turner 3,*
Responses to reviewers’ comments (In blue. Line numbers refer to the version of the revised manuscript with all changes tracked). We have also added responses to the comments and suggestions the reviewers have made on annotated (pdf) copies of the manuscript – attached.
Reviewer 2
I welcomed the opportunity to review the paper 'Using Paleomagnetic techniques to date indigenous archaeological sites in New Zealand' by Poojary et al.
Overall the manuscript is well organized and written and after a set of comments are addressed I think that it is ready to be published.
Before that stage, I would like to view the cited figures which are not appended in the main manuscript and this is a major concern.
I’m really not sure what you mean here! All cited figures (1-10), tables (1, 2) and the appendix were submitted and, I assume, available to the reviewers. We do not propose to append supplementary electronic information.
I suggest to merge figures 1 with 5 (deleting figure 4) I also suggest to merge figures 2 and 3 as one location map, increasing the font sizes and adding the Geographic north and scales for reference.
I’m afraid we do not have time before the allocated deadline to make major changes to Figures as suggested here. The Figures can be scaled according to the amount of detail each contains, so is there any benefit in merging figures which were originally designed to be separate?
I think the font sizes on Figure 2 are OK.
Figure 3a is a historical document, and, as explained above, we have now included the highest resolution copy available. We have also explained the pertinent features in the figure caption and text.
The grids in Figures 2 and 3(b) are oriented to geographic and grid north (the difference is insignificant for the purpose of these figures). Latitude and longitude serve as a scale in Figure 2. We have added a note that grid squares are 1 x 1 km in Figure 3b.
In Figure 6, please add stereonets and Magnetization/ Temperature or mT to discuss the demagnetization trend, and stereonet to discuss if any anisotropy exists and/ or parasitic field is acquired during demagnetization.
We have not shown the demagnetization data on stereonets in Figure 6, since in general the remanent directions change so little that there will be insignificant movement on the stereoplot. All relevant information is more clearly shown by the combination of vector component (Zijderveld) plot and intensity plot (already plotted). We have added temperature and AF steps to the vector component plots (vertical component).
Anisotropy/directional fabric is a feature of the pottery fabrication process - it is not generally a problem with naturally formed stones. Specimens are placed in the oven in alternating orientations between successive demagnetization steps to compensate for any undetected residual field. We regularly check for residual fields in our thermal demagnetizing oven. We have added a note explaining this in the methods section (lines 173-176).
Please add uncertainties to all data points in figure 7 and mark ones which are not deemed reliable.
We have considered this, and conclude that adding MAD circles would clutter the figures unreasonably. As stated in the caption, only those data used to calculate the final ChRM directions are plotted. Discarded data are not shown. We have directed the reader to the Appendix for the full data, including MADs for each specimen ChRM.
Please merge figures 9 and 10.
Can do, but is there any benefit?
After all these changes, please update the manuscript accordingly.
All other major and minor comments regarding manuscript organization, reference citations and discussion of the data and the results can be found in the attached PDF version of the manuscript.
Best wishes
